# Minimal Mechanisms Responsible for the Dispersive Behavior of the Madden–Julian Oscillation

**Kartheek Mamidi** and **Vincent Mathew** *

Department of Physics, Central University of Kerala, Kasaragod 671316, Kerala, India; kartheek@cukerala.ac.in
* Correspondence: vincent@cukerala.ac.in

**Abstract:** An attempt has been made to explore the relative contributions of moisture feedback processes on tropical intraseasonal oscillation or Madden–Julian Oscillation (MJO). We focused on moisture feedback processes, including evaporation wind feedback (EWF) and moisture convergence feedback (MCF), which integrate the mechanisms of convective interactions into the tropical atmosphere. The dynamical framework considered here is a moisture-coupled, single-layer linear shallow-water model on an equatorial beta-plane with zonal momentum damping. With this approach, we aimed to recognize the minimal physical mechanisms responsible for the existence of the essential dispersive characteristics of the MJO, including its eastward propagation ($k > 0$), the planetary-scale (small zonal wavenumbers) instability, and the slow phase speed of about $\approx 5$ m/s. Furthermore, we extended our study to determine each feedback mechanism's influence on the simulated eastward dispersive mode. Our model emphasized that the MJO-like eastward mode is a possible outcome of the combined effect of moisture feedback processes without requiring additional complex mechanisms such as cloud radiative feedback and boundary layer dynamics. The results substantiate the importance of EWF as a primary energy source for developing an eastward moisture mode with a planter-scale instability. The eastward moisture mode exhibits the highest growth rate at the largest wavelengths and is also sensitive to the strength of the EWF, showing a significant increase in the growth rate with the increasing strength of the EWF; however, the eastward moisture mode remains unstable at planetary-scale wavelengths. Moreover, our model endorses that the MCF alone could not produce instability without surface fluxes, although it has a significant role in developing deep convection. It was found that the MCF exhibits a damping mechanism by regulating the frequency and growth rate of the eastward moisture mode at shorter wavelengths.

**Keywords:** Madden–Julian Oscillation; shallow-water model; moisture interactions

## 1. Introduction

Low-frequency moisture modes or intraseasonal oscillations (ISO) in the tropical atmosphere refer to persistent and coherent patterns of atmospheric variability that occur on a typical time scale of around 60–120 days [1]. These modes are characterized by fluctuations in atmospheric circulation and precipitation patterns that can substantially impact regional and global climate. The Madden–Julian Oscillation (MJO), for example, is a large-scale ISO characterized by a pattern of the enhanced and suppressed convection and rainfall that moves across the tropical Indian Ocean and Western Pacific on a time scale of about 30–60 days [2,3]. The MJO affects the distribution and transport of moisture in the atmosphere and can significantly impact monsoon systems and other precipitation patterns.

Modern satellite technology could provide spectral analysis results, which reveal the strong connection between large-scale tropical motions and moist convection [4]. The linear dispersion relation of these tropical modes was successfully derived more than half a century ago by Matsuno [5], using a "classical dry" linear equatorial beta-plane shallow-water model. However, Matsuno's "dry" theory cannot deduce the observed moist tropical modes with lower phase speeds. With the help of normalized power spectra of satellite

data, such as outgoing long-wave radiation (OLR), Kiladis et al. [4] showed that the family of linear wave solutions predicted by Matsuno was characterized by deep convection in the tropics. Hence, these observed modes are later referred to as convectively coupled equatorial waves (CCEWs).

In recent decades, there has been significant research on the dynamics, structure, and propagation characteristics of CCEWs in observation and theoretical studies. Aside from CCEWs, Hendon and Salby [6] and Kiladis et al. [7] observed a dominant mode of tropical variability in the wavenumber–frequency spectra of OLR, concentrated on the intraseasonal timescale and the planetary zonal scale and this dominant signal is referred to as MJO. However, unlike other tropical modes, the MJO does not fall along Matsuno's classical dispersion curve and has a peculiar dispersion, i.e., frequency is nearly constant with the increasing positive zonal wavenumber in a low-frequency regime. The MJO is an organized, eastward propagating ($\simeq 5$ m/s), large-scale convective envelope, and it is embedded with both east and westward-moving convective fine-structures, making it a complex phenomenon to understand [8,9]. Due to its multi-scale moisture-driven structure, it impacts the local and global climatic variabilities and provides a significant source of predictability at intraseasonal scales [10]. In recent decades, through observational research, scientists evaluated the impact of MJO on phenomena ranging from diurnal convective events to large-scale motions such as monsoon variability, tropical cyclones, and extra-tropical circulations [11–13].

Since its first observation, MJO has inspired many researchers to develop a theory for representing its fundamental dynamics [14]. Some scientists have attempted to explain the MJO as an additional eastward propagating moisture mode of the tropical wave spectrum and addressed its underlying physical mechanisms. These theories have relied on Matsuno's original representation of classical dry trapped waves or the same approach with an additional set of equations by including moisture as a new variable in linear and nonlinear regimes. This school of theories includes moisture-mode theories [15–19], trio-interaction theories [20], skeleton theory [13], WISHE-moisture mode theories [21–24], as well as a recent theory by Rostami et al. [25], in which the author critically advocates the significance of moist convection and arrives at an MJO solution as a nonlinear eastward-moving "hybrid structure". However, another school of theories also exists in which the authors explained the MJO dynamics through various physical mechanisms without using moisture as an additional variable, such as gravity wave theory by Yang and Ingersoll [26], the nonlinear solitary wave theory of Yano and Tribbia [27], and the large-scale vortex theory by Hayashi and Itoh [28]. In isolation with the above-mentioned theories, Kim and Zhang [29] recently proposed a simple theoretical framework by revisiting the earlier hypothesis of Chang [30], i.e., introducing viscous damping leads to slow down the Kelvin wave and show the similar signatures of MJO. Kim and Zhang [29] successfully derived an MJO solution with momentum damping alone to the linear equatorial shallow-water equations; however, the authors also suggested that Rayleigh damping could be the possible outcome of moist convective processes.

The two-way interaction between the convection and moisture dynamics is the non-trivial and most debated problem in the field of tropical atmospheric research since the development of the conditional instability of the second kind (CISK) mechanism by Charney and Eliassen [31] and Ooyama [32]. In one way, the moisture dynamics in the tropical atmosphere play a key role in shaping convective interactions. Conversely, convective dynamics can help sustain and amplify low-frequency ISOs by influencing the distribution and transport of moisture, heat, and energy in the atmosphere. In CISK theory, they established that convection due to latent heat release leads to the development of large-scale motion when there is ambient conditional instability. However, Emanuel et al. [33] contradicted the representation of convection in the earlier CISK mechanism, stating that convection acts as a damping source to sustain the large-scale instability and coined the term moist convective damping in their wind-induced surface heat exchange (WISHE) theory.

In tune with the aforementioned linear CISK theory, Lindzen [34] developed a Wave-CISK by emphasizing the role of latent heat release, but the obtained linear solution suffered

from the ultraviolet catastrophe of the largest growth at the smallest scales. Later on, Moskowitz and Bretherton [35] rectified this unrealistic solution by including frictional convergence to the wave-CISK mechanism. However, in their earlier works, this theory was continuously criticized by Emanuel [21] for the requirement of background conditional instability and the treatment of the convective dynamics [36,37]. Meanwhile, ref. [21] developed a new paradigm, in which large-scale motions in the tropics were driven by an energy source known as WISHE or evaporation-wind feedback (EWF). Furthermore, Sobel and Maloney [17], Khairoutdinov and Emanuel [24], Fuchs-Stone and Emanuel [38] successfully explored the role of evaporation fluxes as a primary mechanism in MJO dynamics. In the earlier linear analysis of Fuchs and Raymond [22], they initially thought that MJO is an outcome of the combined effect of cloud radiation feedback (CRF) and WISHE. Later, they established that CRF shows a destabilizing mechanism at the same time, and WISHE develops a primary instability [23]. However, the interaction between radiation and water vapor goes back to the earlier hypothesis of Hu and Randall [39]. Furthermore, Adames and Kim [18] extended this mechanism by modifying the CRF as a function of the wavenumber in their moisture mode theory and established the importance of CRF on MJO dynamics. The role of CRF has also been explored in other theoretical frameworks, such as trio-interaction models by [20] with the inclusion of planetary boundary layer (PBL) dynamics. In this formulation, the authors concluded that the combination of the CRF and PBL leads to the development of planetary scale instability, and the phase speed and growth rates are influenced by the CRF mechanism [40].

Despite the fact that there is a large and diversified pool of theoretical works to draw from, numerous questions remain unsolved (for a complete overview, readers can refer to Zhang et al. [14]). As a result, understanding the MJO's dynamics remains a challenge and is considered the "Holy Grail" of tropical atmospheric research [15]. However, the primary goal of this work is not to answer all of the unanswered questions about MJO dynamics. Instead, we intend to evaluate the moisture feedback processes as a viable physical mechanism responsible for recreating the MJO's dispersive signature. According to Lindzen [34] and Nakazawa [41], low-level moisture convergence is an essential physical process within the tropics, which plays a crucial role in shaping convective dynamics and further influences the large-scale dynamics of low-frequency tropical modes. Therefore, we aim to explore the possibility of visualizing the MJO as an intrinsic mode introduced by moisture feedback processes while encompassing the significance of evaporation-wind feedback and low-level moisture convergence in the tropical atmosphere. The individual roles of the above mechanisms have been extensively demonstrated in earlier theories. However, the possible role of each mechanism or the combination of several mechanisms in determining the dispersive signature of the MJO-like mode remains elusive. Here, we established the mathematical description, which estimates each mechanism's possible role and strength, as well as their possible roles and strengths in combination with one another in exhibiting the MJO-like mode.

This paper is organized as follows. Assuming a simple first baroclinic structure, in the next Section 2, we introduced the system of equations, which describes the dynamics of the tropical waveguide, and we proceed to a linear analysis. We explored the dispersion relation and associated moisture mode characteristics. The results are discussed in Section 3. Finally, the concluding remarks are given in Section 4.

## 2. Theoretical Description

The theoretical framework presented here represents the low-frequency intraseasonal oscillations in the tropical atmosphere. It describes the horizontal structure of the first baroclinic mode in the tropics. It consists of the simplified parameterizations of evaporation-wind feedback and precipitation heating in a nearly saturated convective tropical atmosphere [42]. These approximations are applied to rotating shallow water equations in an equatorial $\beta$- plane, along with an equation for the depth-integrated moisture variable. The complete, dimensional form of the described model is as follows:

$$\frac{DV}{Dt} + f \times V = \nabla\phi - \epsilon V \tag{1}$$

$$\frac{\partial\phi}{\partial t} - \nabla \cdot (\phi V) = Q - \mu\phi \tag{2}$$

$$\frac{\partial q}{\partial t} + \nabla \cdot (Vq) = E - P \tag{3}$$

Equation (1) represents the zonal and meridional momentum equations with linear mechanical damping, where $V(u, v)$ represent the free-tropospheric low-level zonal and meridional velocity perturbations. Equation (2) is derived from continuity, hydrostatic and thermodynamic equations with $\phi$ as a mid-level mean potential temperature, where $\mu$ is the Rayleigh friction and $Q$ is the heating rate due to condensation. Equation (3) is the vertically integrated moisture equation, which describes the evolution of the moisture variable $q$, $E$ represents the evaporation, and $P$ represents the precipitational heating.

The precipitational heating ($P$) is proportional to the column-integrated moisture ($q$). It is associated with the diabetic heating of the deep convection and governed by the moisture relaxation time ($\tau$). According to Bretherton et al. [43], the expression is as follows:

$$P = \frac{q - \bar{q}}{\tau}; \quad q = \bar{q}(1 + s) \tag{4}$$

where $s$ describes the perturbation over equilibrium value $\bar{q}$ over $q$.

We adopted the simplified treatment of parameterization schemes for a heating rate similar to previous moist models [44], which are given by

$$Q = \eta P \tag{5}$$

where

$$\eta = \frac{L_v}{\rho_a C_p H}$$

Here, $L_v$ is the latent heat of condensation, $\rho_a$ is the density of air, and $C_p$ is the specific heat at constant pressure.

The surface evaporation is parameterized by the simplified bulk-aerodynamic formula [45,46], and given by

$$E = -\rho_a C_D \Delta\bar{q}_s.|\vec{u}| = -\Lambda u \tag{6}$$

where $C_D$ is the drag coefficient, $\Delta\bar{q}_s$ (=3 g kg$^{-1}$) is the saturation relative humidity difference between the sea surface and the anemometer level, and $|\vec{u}|$ represents the direction of mean zonal winds, whilst the parameter $\Lambda$ in Equation (6) represents the EWF, and is considered positive for mean surface easterlies and negative for mean westerlies. We intend to find instabilities specific to planetary-scale wavelengths with our framework, similar to the WISHE-moisture mode theory of Fuchs and Raymond [23]. Therefore, as suggested by Emanuel [21] and Neelin et al. [45], we assumed easterly global mean zonal winds to be average zonal winds in the tropics [47–49]. The WISHE mechanism, specifically the choice of global mean easterly wind in simulating MJO-mode, is questioned by various other theories due to the presence of strong mean westerlies over warm pool regions (western Pacific and Indian oceans) [50,51]. However, the choice of the global mean wind is justified by the linearity of our model since we seek linear solutions that are unique to the global scale ($k = 1$).

Considering that the primary objective of our study is to investigate the linear modes in a low-frequency regime, we linearized Equations (1)–(3) and used the long-wave approximation. We further neglected the zonal moisture advection term in order to focus

on the role of EWF and MCF in linear wave dispersion. Therefore, the reduced governing equations for studying the moist tropical modes are as follows

$$\left(\frac{\partial}{\partial t} + \epsilon^*\right)u - f^*v - \frac{\partial \phi}{\partial x} = 0 \tag{7}$$

$$f^*u - \frac{\partial \phi}{\partial y} = 0 \tag{8}$$

$$\left(\frac{\partial}{\partial t} + \mu^*\right)\phi - \left\{\frac{\partial u}{\partial x} + \frac{\partial v}{\partial y}\right\} - \eta \alpha s = 0 \tag{9}$$

$$\frac{\partial s}{\partial t} + \Gamma_q\left\{\frac{\partial u}{\partial x} + \frac{\partial v}{\partial y}\right\} + \Lambda u + \alpha s = 0 \tag{10}$$

where

$$\Gamma_q = 1 - \left[\frac{H}{\bar{q}}\right]\frac{d\bar{q}}{dz}; \quad f^* = \frac{y}{2}$$

The parameter $\Lambda$ represents the nondimensional values of the evaporation-wind feedback component, estimated from the empirical relation given in Table 1. $\alpha$ is the nondimensional value of the moisture relaxation timescale, $\epsilon^*$ represents the linear mechanical damping parameter, $\mu^*$ is the Rayleigh friction or Newtonian cooling, and $\Gamma_q$ is the moisture convergence feedback. The list of model parameters, including nondimensional parameters, and their values is given in Table 1.

**Table 1.** Description of the model parameters.

| Description | Parameter | Definition/Units | Average Value |
|---|---|---|---|
| Dry gravity wave speed | C | ms$^{-1}$ | 50 |
| Time scale | $T_0$ | $[2C\beta]^{\frac{1}{2}}$ | 8.33 h |
| Length scale | $L_0$ | $[\frac{C}{2\beta}]^{\frac{1}{2}}$ | 40,000 km |
| Meridional gradient of Coriolis parameter | $\beta$ | m$^{-1}$s$^{-1}$ | $2.28 \times 10^{-11}$ |
| Moisture convergence feedback * | $\Gamma_q$ | $1 - (\frac{H}{\bar{q}})\frac{d\bar{q}}{dz}$ | $-0.05$ |
| Convective time lag * | $\alpha$ | $[\frac{T_0}{\tau}]$ | 0.25 |
| Rayleigh damping * | $\epsilon^*$ | $[\frac{T_0}{T_d}]$ | 0.04 |
| Evaporation-wind feedback * | $\Lambda$ | $\rho C_D \Delta \bar{q}_s \frac{T_0}{q}$ | 0.05 |

* for nondimensional parameters.

*Dispersion Relation*

Equations (7)–(10) represent a closed system of linear partial differential equations with constant coefficients. We assume that solutions have a zonally propagating plane-wave structure of the form $\zeta(x, y, t) = \zeta(y)e^{[i(kx - \omega t)]}$, where $\zeta$ represents any of the perturbed variables $u, v, \theta$, or $s$. "$k$" is the zonal wavenumber $k = 1, 2, 3, \ldots$ and $\omega$ is the complex frequency. The phase speed and growth rate can be deduced by a real part $Re(\omega)/k$ and imaginary parts $Im(\omega)$ of the frequency $\omega$. For simplicity, we considered the time scale for mechanical damping and Newtonian cooling to be the same ($\epsilon^* = \mu^* = \epsilon$). Therefore, we can write the linear system of equations in terms of frequency and wavenumber as

$$\sigma u - if^*v + k\phi = 0 \tag{11}$$

$$f^*u - \frac{d\phi}{dy} = 0 \tag{12}$$

$$\sigma\phi + ku - i\frac{dv}{dy} - i\eta\alpha s = 0 \tag{13}$$

$$\omega^* s - [k\Gamma_q - i\Lambda]u + i\Gamma_q\frac{dv}{dy} = 0 \tag{14}$$

with

$$\sigma = \omega + i\epsilon; \quad \omega^* = \omega + i\alpha$$

By eliminating the variable $s$ from Equations (13) and (14), we can achieve a new system of equations, whilst further conducting the mathematical manipulation and eliminating the variable $\phi$ results in a second-order ordinary differential equation in $v(y)$.

$$\frac{d^2v}{dy^2} + a_0y\frac{dv}{dy} + a_1v = 0 \tag{15}$$

while the parameters

$$a_0 = \frac{\bar{\Lambda}}{\sigma\Gamma}; \qquad a_1 = -\left[\frac{\bar{\Gamma}}{\sigma\Gamma} + \frac{1}{\Gamma}\right]$$

Furthermore, by applying the variable transformation, we can achieve a reduced second-order differential equation in $V(\psi)$. This is similar to Matsuno's equation, where

$$\frac{d^2V}{d\psi^2} + (C_2 - C_1\psi^2)V = 0 \tag{16}$$

The coefficients $C_1$ and $C_2$ are

$$C_1 = \frac{1}{\Gamma} - \left[\frac{\bar{\Lambda}}{2\sigma\Gamma}\right]^2; \quad C_2 = -\left[\frac{\bar{\Gamma}}{\sigma\Gamma} + \frac{\bar{\Lambda}}{2\sigma\Gamma}\right]$$

where

$$\bar{\Gamma} = \Gamma k - \bar{\Lambda};$$

$$\bar{\Lambda} = \frac{\eta\alpha\Lambda}{\omega^*}; \quad \Gamma = 1 - \frac{i\eta\alpha\Gamma_q}{\omega^*}$$

However, we assume a wave motion near to the equatorial region and we apply the boundary conditions such as $V \to 0$, when $\psi \to \pm\infty$. With these boundary conditions, the solution to the above Hermite differential equation can be either of the form

$$V = 0; \text{or} \quad V(\psi) = Ce^{-c_ny^2}H_n(\psi) \tag{17}$$

where $H_n(\psi)$ is the nth-order Hermite polynomial. The solution given by Equation (17) is only possible when it satisfies the following relation

$$C_2C_1^{-1/2} = 2n + 1; \quad n = -1, 0, 1, 2, \ldots \tag{18}$$

where $n$ is an integer, which represents the meridional mode.

Solving Equation (18) results in the dispersion relationship, which can be written as a characteristic polynomial in $\omega$ as follows

$$N^2\omega^4 - a_0\omega^3 - a_1\omega^2 - a_2\omega + a_3 = 0 \tag{19}$$

where the coefficients $a_0$, $a_1$ $a_2$ $a_3$ are as follows,

$$a_0 = iN^2[\alpha(1 + \Delta) - 2\epsilon],$$

$$a_1 = [k^2 + N^2(\epsilon^2 + \alpha^2 + 4\epsilon\alpha - 2\epsilon\eta\alpha\Gamma_q - \eta\alpha^2\Gamma_q)],$$

$$a_2 = 2i\alpha k^2 \Delta - \eta\alpha\Lambda k + iN^2[\epsilon^2\alpha(1+\Delta) + 2\epsilon\alpha^2\Delta],$$

$$a_3 = \alpha^2 k^2[1 - 2\eta\Gamma_q + \eta^2\Gamma_q] + i\eta\alpha^2\Lambda k\Delta - \frac{(1+N^2)}{4}\eta^2\alpha^2\Lambda^2 + N^2\epsilon^2\alpha^2\Delta$$

with

$$N = 2n+1; \quad \Delta = 1 - \eta\Gamma_q$$

## 3. Results and Discussion

Before examining the complete dispersion relation in detail, we consider the limiting case with no feedback processes, i.e., neglecting the precipitational heating, moisture convergence ($\Gamma_q = 1$), evaporation-wind feedback ($\Lambda = 0$), and mechanical damping ($\epsilon = 0$). This approximation results in reduced classic Matsuno's dispersion Equation ($\omega^2 - k^2 = 0$) for a special case of zero meridional wind ($v = 0$). This dispersion relation illustrates that, in the absence of feedback processes, the solution exhibits a nondispersive eastward propagation, and this mode's behavior closely resembles the pattern followed by the dry Kelvin mode of Matsuno [5]. Hence, we argue that the additional physical mechanisms, such as moisture feedback processes (EWF and MCF), which could provide an energy source, are required to develop an eastward instability at the planetary scale.

We further studied the specific solutions of Equation (19) using the standard parameter set detailed in Table 1. The solutions are shown in Figure 1a, where the real part of frequency ($Re(\omega)$) is plotted against the zonal wavenumber ($k$), and the corresponding growth rate is shown in Figure 1b. We obtained two modes of the solution in the low-frequency domain: one propagating in the westward direction ($k < 0$); and the other in the eastward direction ($k > 0$). The observed wave mode closest to the modeled westward branch in the low-frequency regime is the $n = 1$ traditional moist Equatorial Rossby wave. The additional low-frequency eastward moisture mode is unstable at planetary-scale zonal wavenumbers ($k \leq 2$) with a frequency that is weakly dependent on the zonal wavenumber. Both the low-frequency modes show the highest growth rate at longer wavelengths ($k < 2$) and slowly fall at shorter wavelengths ($k > 4$). However, the eastward mode shows the maximum growth rate compared to the westward mode. Hence, from this illustration, it is clear that the manifestation of the eastward low-frequency mode is due to the existence of moisture feedback processes.

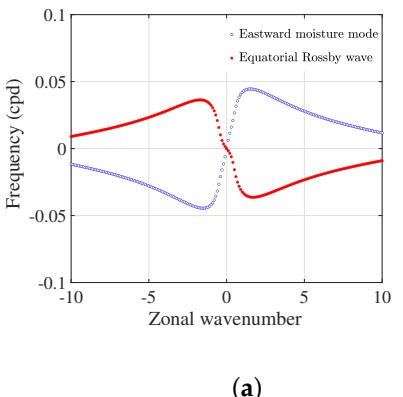
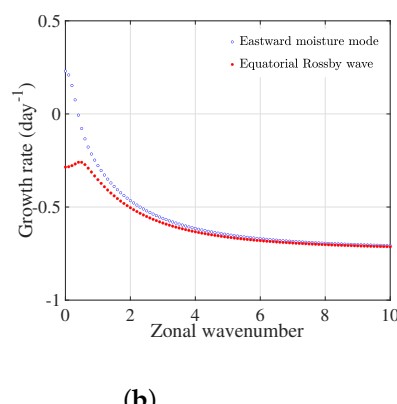

(**a**)                    (**b**)

**Figure 1.** The low-frequency modes of $\omega(k)$ versus $k$ from Equation (19): when $\alpha = 0.25$, $\Gamma_q = -0.05$, $\Lambda = 0.05$, and $\epsilon = 0.04$. (**a**) Frequency (cycles per day) of eastward mode (blue open circles) and westward mode (red dots); (**b**) Growth rate (day$^{-1}$) of eastward mode (blue open circles) and westward mode (red dots).

The dynamic features of the eastward mode are shown in Figure 2. The new eastward propagating mode has a period within the intraseasonal range (Figure 2a). It is an unstable mode with the maximum growth rate at longer wavelengths, indicating a preferred planetary-

scale unstable mode (Figure 2b), and the dimensional phase speed at the planetary-scale wavenumber is nearly 5 m/s (Figure 2c). Similarly to the trio-interaction theory of the MJO proposed by Wang et al. [20], this model fundamentally deals with the moisture feedback processes. However, we did not include boundary-layer frictional convergence; instead, we assumed that the precipitational heating is linearly proportional to a column moisture anomaly, and therefore, we considered the free tropospheric moisture convergence. While the two processes crucial to the WISHE-moisture model [23] and boundary layer quasi-equilibrium (BLQE) theory [24] are surface enthalpy fluxes and cloud-radiation effects, we did not involve CRF feedback and boundary layer in our formalization. However, we incorporated a similar framework of the WISHE mechanism in this model by considering the mean easterlies as background winds. The other class of moisture-mode theory is based on the weak temperature gradient approximation (WTG) [18,51], in which the moisture is retained as the only prognostic variable, and the simulated MJO constitutes the moisture-mode by considering the mean westerlies (emphasizing the warm pool region). Sobel and Maloney [52] and Adames and Kim [18] combined the effects of surface evaporation and zonal moisture advection, and they explained the effect of the westerly WISHE mechanism and eastward propagation of the obtained moisture mode. However, we did not consider the background zonal moisture gradients in the moisture budget equation to reduce the analytical complexity. Despite these reductions in physical processes, the distinct dispersive signature produced by our framework is consistent with the dispersive characteristics of the MJO as a linear solution simulated by various theoretical studies such as Wang et al. [20], Fuchs and Raymond [23] as well as Wang and Sobel [53]. In particular, our model's eastward moisture mode solution is similar to the linear solutions of WISHE-moisture mode theory and BLQE theory with a zero meridional wind ($v = 0$) approximation of Fuchs and Raymond [23], Fuchs-Stone [54], and similar to the $n = 1$ mode of Fuchs-Stone and Emanuel [38], Emanuel [55] with a meridional flow.

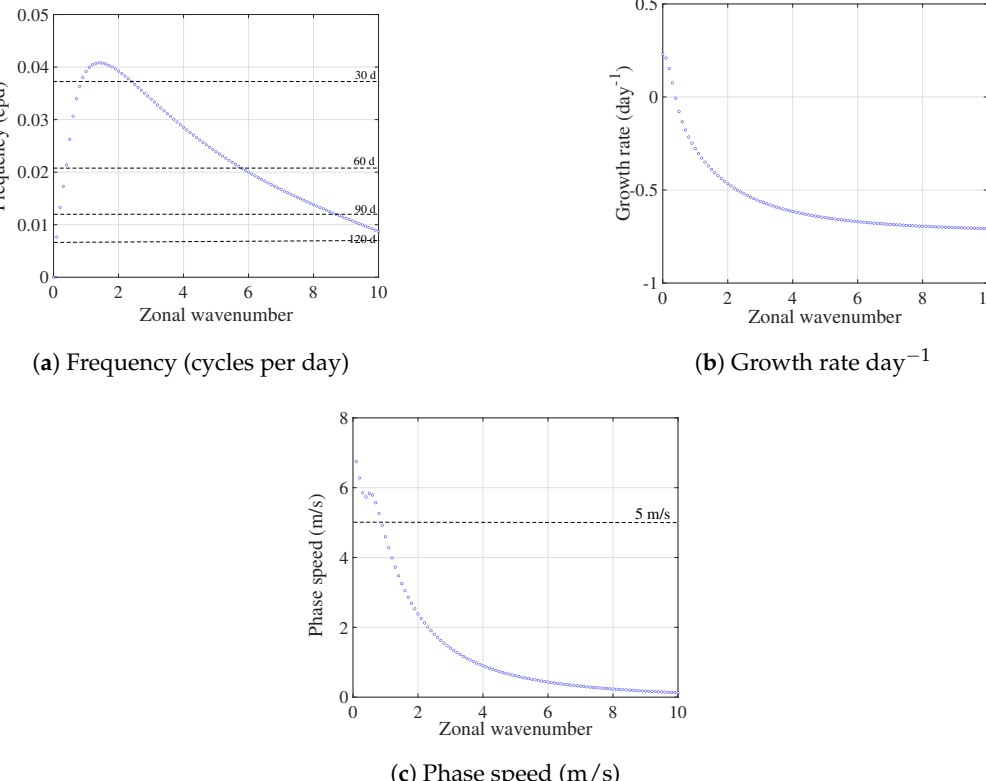

(**a**) Frequency (cycles per day)

(**b**) Growth rate day$^{-1}$

(**c**) Phase speed (m/s)

**Figure 2.** The eastward propagating moisture mode: when $\alpha = 0.25$, $\Gamma_q = -0.05$, $\Lambda = 0.05$, and $\epsilon = 0.04$.

From Figure 2, it is clear that the existence of the eastward moisture mode and its planetary-scale instability is due to the competing mechanism between the moisture

feedback processes. Therefore, we intend to identify the individual roles of the MCF and EWF. We first exclude the EWF (by setting Λ to zero) and by invoking the MCF alone. This approximation results in an unrealistic solution, as shown in Figure 3a (blue open circles), which explains that there is no energy source for an eastward moisture mode to grow when the EWF is absent (Λ = 0), even though MCF is active. This result agrees with earlier theoretical works based on the WISHE mechanism. Furthermore, we varied the EWF values by holding the other parameter constant, which resulted in a modified eastward mode with respect to EWF, illustrated in Figure 3. It shows that, for higher values of EWF, such as Λ = 1.5 and 3, the obtained mode retains a typical instability and exhibits nonlinearity at planetary-scale wavelengths. However, the frequencies are strongly affected by the strength of the EWF parameter, showing a significant increase in frequency with an increase in EWF strength, retaining the instability at planetary-scale ranges. Similarly, the growth rate curves show maximum growth at planetary-scale wavelengths ($k > 3$) for increasing the Λ values. However, the modified mode is dampened for intermediate and shorter wavelengths (Figure 3b). Increasing the EWF values up to 1 (Λ < 1) does not influence the fundamental behavior of the eastward moisture mode, i.e., the phase speed decreases with the increasing wavenumber for Λ < 1, with a maximum phase speed at longer wavelengths. However, the Λ values above 1.5 exhibit a very slow phase speed in longer wavelength ranges $k < 1.5$, further attaining a maximum phase speed at $k > 1.5$ and slowly decreasing with the wavenumber (Figure 3c). This typical behavior explains the nonlinear nature of the mode at longer wavelengths. Therefore, it is clear that the EWF mechanism is solely responsible for the development of planetary-scale instability in our model.

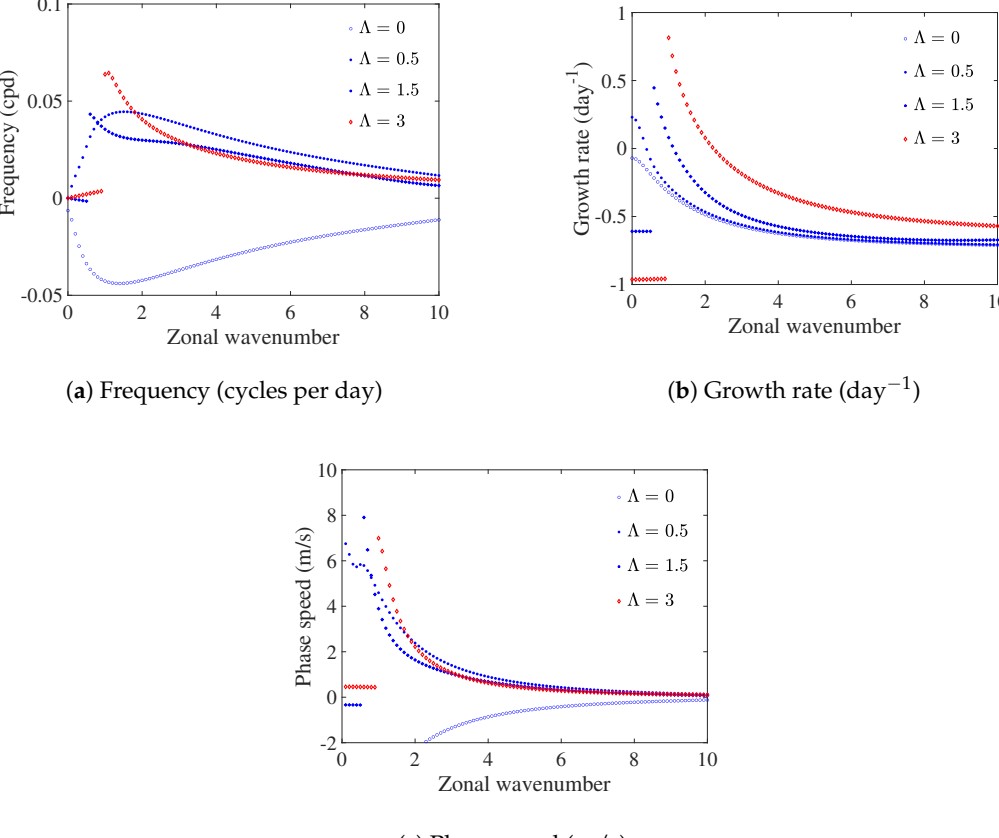

(**a**) Frequency (cycles per day)

(**b**) Growth rate (day$^{-1}$)

(**c**) Phase speed (m/s)

**Figure 3.** The role of the feedback parameter EWF on the eastward propagating moisture mode (Λ = 0, 0.5, 1.5, and 3); other parameters are fixed ($\alpha$ = 0.25, $\Gamma_q$ = −0.05).

Furthermore, we explored the role of MCF in the development of planetary-scale instability and propagation of the eastward moisture mode. For identifying the physical

mechanism of MCF alone, we considered two possible cases: first, the moist-neutral case ($\Gamma_q = 0$); and second, the case when the moisture convergence feedback is off ($\Gamma_q = 1$) while retaining the constant values of EWF ($\Lambda = 0.05$) and MRT ($\alpha = 0.33$). Figure 4a depicts the modified eastward moisture mode due to different $\Gamma_q$ values. From this illustration, it is clear that the instability remains the same at planetary-scale wavelengths regardless of whether the system is in a moist-neutral case or without MCF. However, the frequency is modified in such a way that, for a moist-neutral case, it is in the intraseasonal frequency range. Without a moist convergence feedback case, the eastward moisture mode's frequency is reduced, and it is not in the range of an observational time period. Corresponding growth rate curves are shown in Figure 4b. These curves demonstrate that, for the moist-neutral case, the growth rate of the eastward moisture mode slowly decreases with the wavenumber and becomes dampened at shorter wavelengths. Furthermore, the growth rate for the case without MCF is higher than the prior case but shows a similar qualitative nature. The phase speed curves are shown in Figure 4c, and these results show that a maximum phase speed is observed at the planetary-scale wavelength when the moist-neutral case is invoked. However, for the other case, it shows a very slow phase speed at similar wavelengths. Furthermore, we analyzed the sensitivity of MCF by varying its value to $\Gamma_q = -0.5$ and $-1$. In comparison with the EWF, the MCF has negligible effects on the development of planetary-scale instability (Figure 4a). However, there is a significant effect on the dispersion relation at smaller wavelengths. It is clear that, at shorter wavelengths, the obtained mode becomes more damped, depending on the MCF value. Hence, MCF mainly acts to regulate the frequency of the eastward moisture mode. We can observe the growth rate pattern of the modified eastward moisture mode ($\Gamma_q$ values $-0.5$ and $-1$) from Figure 4b. The growth rate shows a peculiar behavior; it falls sharply with the wavenumber up to the planetary scale range and then increases for intermediate wavenumbers. Furthermore, it becomes neutral for higher wavenumbers. Therefore, from this observation, it is clear that the MCF does not favor the planetary scale instability because the MCF-induced growth rate does not favor the longer wavelength but is rather significant at shorter wavelengths. Similarly, variations in the phase speed at longer wavelengths are not observed, but at shorter wavelengths, it rapidly decreases and becomes unrealistic. The results demonstrate that either of the feedback processes alone cannot establish the MJO-like signature. Although EWF introduces the eastward perturbation, it cannot reduce the frequency of the eastward moisture mode at higher zonal wavenumbers, but the MCF regulates the frequency of the eastward moisture modes at large zonal wavenumbers.

Finally, we explored the role of another significant model parameter, i.e., the moisture relaxation time-scale $\tau$ ($\alpha^-$), which determines the dynamics of convective interactions through precipitational heating in the tropical atmosphere. We initially chose this to be one day based on physical arguments, and recent numerical model simulations by [56]. At this timescale, along with the active EWF and MCF, we simulated an unstable eastward moisture mode Figure 5. Furthermore, we were interested in identifying the behavior of the eastward moisture mode at different MRTs, as shown in Figure 5a. For a short MRT ($\tau = 2$ h), the frequency of the solution increases with the wavenumber and behaves like a moist-Kelvin wave. The growth rate decreases sharply with the wavenumber and becomes further damped at smaller wavelengths Figure 5b. The phase speed reaches the maximum value ($\approx 15$ m/s) at the planetary-scale wavenumber. A similar dispersion behavior is observed when MRT increases to 12 h. The increase in MRT to 1.5 days results in a decrease in frequency in the intraseasonal range, with a peculiar dispersion relationship similar to the observed MJO-mode. In this case, the solution attains a maximum growth rate at longer wavelengths, decreases slowly with the wavenumber at the intermediate scale, and finally becomes stable for shorter wavelengths. Moreover, the phase speed is also in the range of 5–6 m/s at planetary scale zonal wavenumbers. Furthermore, increasing MRT values to 2.5 days decreases the frequency with an unrealistic phase speed and growth rate. These results agree with the existing models based on the trio-interaction theory by [20]. However,

with a convective adjustment scale of 12 h, their simulated moisture mode results from a complex interaction among boundary layer convergence, CRF, and moisture processes.

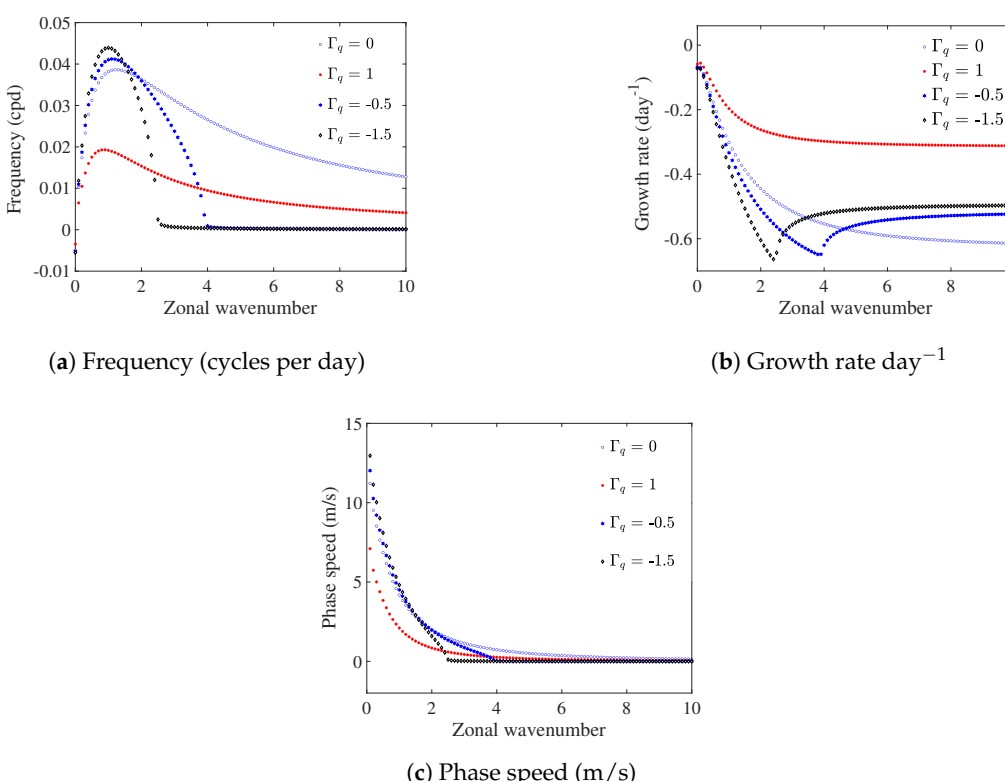

(**a**) Frequency (cycles per day)　　　　　　　　　　　(**b**) Growth rate day$^{-1}$

(**c**) Phase speed (m/s)

**Figure 4.** Role of MCF on the eastward propagating moisture mode ($\Gamma_q = 0$, 1, $-0.5$, and $-1.5$); other parameters are fixed ($\alpha = 0.25$, $\Lambda = 0.05$).

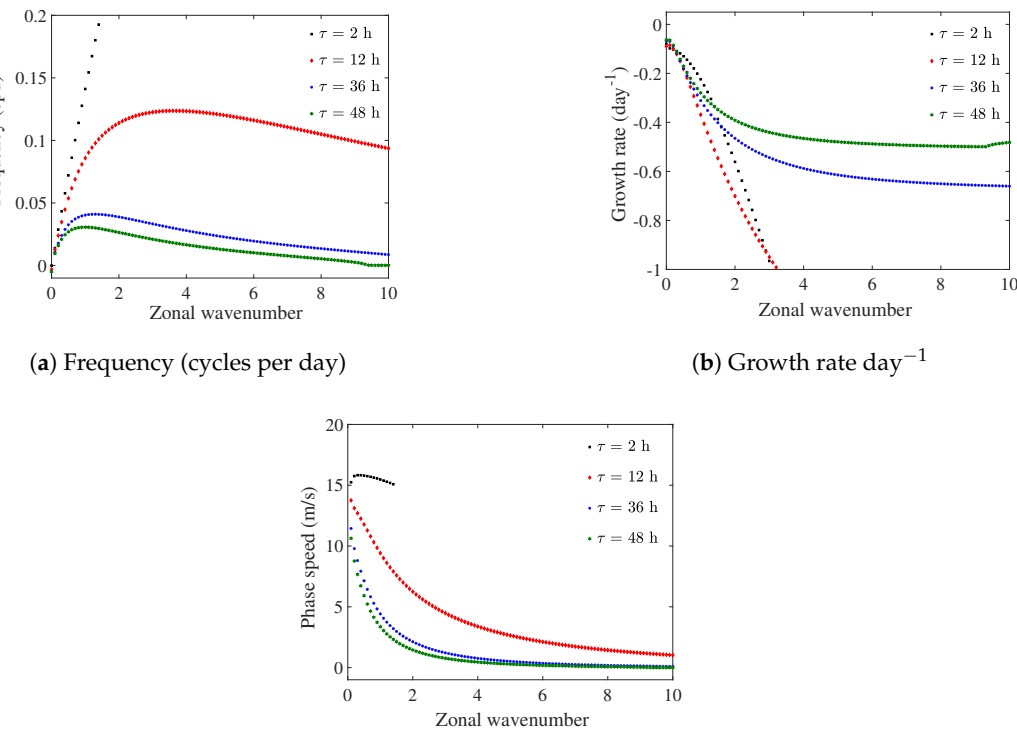

(**a**) Frequency (cycles per day)　　　　　　　　　　　(**b**) Growth rate day$^{-1}$

(**c**) Phase speed (m/s)

**Figure 5.** Role of MRT on eastward propagating moisture mode ($\tau = 2$ h, 12 h, 36 h, and 48 h); other parameters are fixed ($\Lambda = 0.5$, $\Gamma_q = -0.05$).

## 4. Conclusions

In the present work, we investigated the influence of the EWF and MCF on the dispersive signature of low-frequency tropical intra-seasonal oscillation using a moisture-coupled, linear shallow-water model on the equatorial beta plane. We considered the moisture budget equation, in which the linear evolution of precipitational heating is proportional to the column-integrated moisture anomaly, including free tropospheric moisture convergence and evaporation-wind as moisture feedback parameters. The global mean easterly winds are assumed to determine the EWF parameter since we seek specific linear solutions at planetary-scale wavelengths, similarly to the WISHE mechanism followed by Fuchs and Raymond [23]. With these two feedback mechanisms, we search for the linear solutions intrinsic to the low-frequency regime of the tropical waveguide. The analytical dispersion relation gives two different unstable moisture modes (Figure 1). The first is the westward moisture mode, closely resembling the traditional $n = 1$ planetary-scale equatorial Rossby wave. The other solution is an eastward perturbation having the largest instability at longer wavelengths. This new moisture mode exhibits a characteristic dispersion behavior on an intra-seasonal time-scale, i.e., its frequency is almost constant as a function of the wavenumber, a slow eastward propagation with a phase speed of around 5–6 m/s, and shows a critical instability at planetary-scale wavelengths. All these dispersive characteristic features agree with the earlier theoretical models [23,53] and are also consistent with the observational wavenumber-frequency power spectrum of Kiladis et al. [4]. Hence, we considered this new eastward mode as an MJO-like mode; however, the additional essential mechanisms, such as the nonlinear effect of moist convection, are required to explain the MJO fundamental features fully, including the quadruple structure, showcasing two-dimensional condensation patterns and the inter-seasonal longevity [25]. The dispersion characteristics of the obtained eastward moisture mode are similar to the linear solutions obtained from established theoretical frameworks, such as WISHE and trio-interaction theory, despite the nature of the model being a reduced moisture model, without the other complex interactions, such as cloud radiation feedback and boundary-layer dynamics.

Furthermore, we systematically scrutinized the influence of the moist-convective feedback processes on this eastward moisture mode, and we verified the necessity of each process in sustaining the planetary-scale instability, regulation in intra-seasonal frequency, and continuous eastward propagation. In this context, we simulated the dispersion relation by switching off the EWF term, and by invoking the MCF alone, this approximation does not yield an eastward moisture mode. Therefore, we conclude that the fundamental source for the existence of the eastward moisture mode and its planetary-scale instability is evaporation wind feedback, similarly to the WISHE-moisture mode. The strength of the EWF parameter strongly influences the frequencies, which signifies that increasing the strength of surface fluxes results in increasing the frequency of the eastward mode. Similarly, the strength of EWF shows significant variations in the modulation of instability; nevertheless, the eastward moisture mode remains unstable at planetary-scale wavelengths. Similarly, the growth rates also increase with the increasing strength of EWF at planetary-scale ranges ($k > 3$). The sensitivity studies of MCF on dispersion relation show that the results of this model suggested that the moisture convergence feedback alone could not produce instability at longer wavelengths; however, it regulates the frequency and growth rates at shorter wavelengths. The results demonstrate that either of the feedback processes alone cannot establish the MJO-like signature and suggest that the MJO-like dispersive signature is due to the competing mechanism of both moisture feedback processes. Similarly to the WISHE theory and the trio-interaction model, our model's simulation is also sensible to the moisture relaxation time-scale, which decides whether the obtained mode is a moist Kelvin mode or a new eastward moisture mode, considered MJO-like mode. With its reduced moisture framework, the present model could deduce the dispersive characteristics of the MJO-like mode. However, the inclusion of nonlinear effects associated with precipitational heating, boundary layer dynamics, strong

mean westerlies, or moisture advection, and other scale interactions are necessary for understanding the complex, multi-scale structure of the MJO.

**Author Contributions:** Conceptualization, K.M.; methodology, K.M. and V.M.; formal analysis, K.M.; investigation, V.M.; writing—original draft preparation, K.M.; writing—review and editing, K.M.; visualization, K.M.; supervision, V.M. All authors have read and agreed to the published version of the manuscript.

**Funding:** This research received no external funding.

**Data Availability Statement:** The paper is theoretical, and no data are used. Moreover, the plots were generated from the equations, as the model is purely analytical.

**Conflicts of Interest:** The authors declare no conflict of interest.

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
