# Peer review of "Minimal Mechanisms Responsible for the Dispersive Behavior of the Madden–Julian Oscillation"

_climate, doi:10.3390/cli11120236_

Round 1
Reviewer 1 Report
In this study, the authors seek to develop a theory of the MJO that can explain the main features of the MJO, such as its scales and slow eastward propagation, with minimal physical mechanisms. The approach they adopted is to expand the classical Matsuno shallow water equations by including terms representing evaporation wind feedback (EWF) or WISHE, moisture convergence feedback (MCF), and linear damping. Their solutions indeed capture the main features of the MJO. But this study suffers from fundamental issues that must be thoroughly addressed before this manuscript can be accepted for publication.
Major Comments
The main claim of this study is that, in the presence of EWF/WISHE and MCF, there is no need for radiation feedback to generate an unstable MJO solution. But this has been shown by Wang et al. (2016) in their trio-interaction theory of the MJO, where radiation feedback helps but is not essential. The main difference is that they rely on boundary-layer frictional convergence to generate instability, whereas this study relies on troposphere moisture convergence (MCF) and EWF/WISHE. This is where the major problem resides:
The assumption of a mean easterly background to make EWF/WISHE work is contradictory to observations, as pointed out long time ago (Wang, 1988). Without a mean easterly background, MJO wind anomalies are stronger west of its convection center rather than to the east, which would make the MJO move westward based on the EWF/WISHE mechanism. This is the most damaging flaw of this study because EWF/WISHE is at the center of this theory but is based on an unjustified assumption. This assumption will have to be removed for this theory to be valid.
In contrary to the objective of this study to seek the minimal physical mechanisms for the main features of the MJO, its theory is much more complicated than the MJO theory by Kim and Zhang (2021), in which no explicit moisture is required as long as sufficient zonal momentum damping is provided. Their theory is far more “bare-bone” than the theory presented in this manuscript. So, this study does not achieve what it is set to.
Given these two main issues, it is difficult to see the novelty of this study and the science it tries to advance. But the authors should be given a chance to address these issue.
Minor Comments
L179: m should be m*.
Last equation of page 5: e, instead of e* should be used to be consistent to the notation afterward.
Figures: It is difficult to distinguish the different lines. Why not use different colors?
What is the sensitivity of the result to the value of Rayleigh damping?
References
Wang, B. (1988b), Comments on ‘‘An air-sea interaction model of intraseasonal oscillation in the tropics,’’ J. Atmos. Sci., 45, 3521–3525.
Wang, B., Liu, F., & Chen, G. (2016). A trio‐interaction theory for Madden–Julian oscillation. Geoscience Letters, 3, 34. https://doi.org/10.1186/s40562-016-0066-z
Kim, J.E. and Zhang, C., 2021. Core dynamics of the MJO. Journal of the Atmospheric Sciences, 78(1), pp.229-248.
Reviewer 2 Report
Dear editor,
The authors aim to illuminate the relative contributions of moisture feedback processes to the tropical intraseasonal oscillation, specifically the Madden-Julian Oscillation (MJO).
Utilizing a reliable methodology, the authors have arrived at significant results. However, the manuscript requires substantial revisions in order to meet the standards for acceptance as a scientific article; the current version is not suitable for publication.
Here are some points regarding the issues or questions raised in the text:
-
In lines 64-68, where the authors elaborate on an alternative array of theories concerning the dynamics of the MJO, the citations should be updated. Recent research by Rostami et al. (2022) and Rostami and Zeitlin (2020) has shown that moist convection serves as an essential requirement for the creation of an MJO-like structure. This structure manifests as a hybrid fusion of the quasi-equatorial modon and the baroclinic Kelvin wave. Thus, there appears to be a convergence of theories regarding the necessity of moist convection.
-
In the Introduction, the authors should also address a recent study by Kim and Zhang (2021). These researchers have recently identified a slowed-down Kelvin-wave (KW) solution that exhibits both intraseasonal periodicity on the planetary scale and eastward propagation, akin to that of the MJO. This was achieved by introducing momentum damping to the linear equatorial shallow-water equations within a "dry" environment. Consequently, similar dispersion relation outcomes could be attained even in a dry scenario through the mechanism of the slowed-down Kelvin wave.
-
Line 142 states that "... P represents the precipitational heating," yet the concept of precipitational heating is not expounded upon within the manuscript.
-
Line 148 indicates that "... s describes the evolution of the perturbation moisture variable." In fact, the governing equation governing the evolution of this variable (s) has been outlined within the manuscript as equation 10, which incorporates strong easterly wind anomalies (Λ). However, it is noteworthy that the authors have demonstrated significant sensitivity of the results to Λ, leading to divergence in the outcomes. Consequently, it is imperative for the authors to expound upon how the results are influenced by Λ in both the abstract and conclusion sections.
-
The definition of Λ in line 156 is inconsistent with that in line 166. This inconsistency makes it challenging for the reader to discern the corresponding meaning of Λ. To enhance clarity, it is necessary to provide a comprehensive definition of Λ in line 156.
-
The Coriolis parameter f* in equations 7 and 8 has not been defined within the manuscript.
-
In fact, Figure 1 (a) indicates that the amplitude of the dispersion relation slope for eastward propagation and Rossby waves is nearly identical, whereas the latter should inherently be higher. This study should address how this phenomenon is explained within its context.
-
Lines 230-231 state, "... Based on the minimum criteria and desirability formulated by Zhang et al. [14], we identify our eastward moisture mode as MJO-like." However, asserting that an eastward propagation is MJO-like merely based on gradual eastward movement falls short. Essential requisites include demonstrating a quadruple structure, showcasing two-dimensional condensation patterns, and establishing inter-seasonal longevity. Consequently, considering a reduction in the certainty of the results as a MJO-like mode would be reasonable.
-
What are the significant differences between this study and other similar studies focusing on moisture modes, such as the WISHE theory, etc.? These dissimilarities have not been emphasized within the manuscript.
-
The superposition of various plots in Figures 1, 3, 4, and 5 lacks differentiation between them. It is necessary to rework these figures using distinct colors, varying line thicknesses, or different sizes, among other possible adjustments, in order to enhance visibility for the reader.
-
In the Conclusion, it is mentioned that "... it can still deduce the fundamental characteristics of intra-seasonal oscillation." This statement lacks precision as "fundamental characteristics" encompass more than just the dispersion relation. The authors should specify the exact outcomes related to growth rate, phase speed, etc., rather than making a broad reference to the fundamental characteristics of the MJO. Consequently, I believe this matter requires careful attention throughout the entirety of the manuscript.
References
Rostami, M., Zhao, B. & Petri, S.(2022) On the genesis and dynamics of Madden–Julian oscillation-like structure formed by equatorial adjustment of localized heating. Quarterly Journal of the Royal Meteorological Society, 148(749), 3788–3813. Available from: https://doi.org/10.1002/qj.4388
Rostami, M, Zeitlin, V. Can geostrophic adjustment of baroclinic disturbances in the tropical atmosphere explain MJO events?. QJR Meteorol Soc. 2020; 146: 3998–4013. https://doi.org/10.1002/qj.3884
Kim, J., and C. Zhang, 2021: Core Dynamics of the MJO. J. Atmos. Sci., 78, 229–248, https://doi.org/10.1175/JAS-D-20-0193.1.
NA
Round 2
Reviewer 2 Report
The authors have effectively addressed the requested changes, and the current version is suitable for publication.
The English language in the text requires minor corrections.